# Impact of Line Edge Roughness on ReRAM Uniformity and Scaling

**DOI:** 10.3390/ma12233972

**Published:** 2019-11-30

**Authors:** Vassilios Constantoudis, George Papavieros, Panagiotis Karakolis, Ali Khiat, Themistoklis Prodromakis, Panagiotis Dimitrakis

**Affiliations:** 1Institute of Nanoscience and Nanotechnology, NCSR Demokritos, 15341 Aghia Paraskevi, Greece; g.papavieros@inn.demokritos.gr (G.P.); p.karakolis@inn.demokritos.gr (P.K.); 2Nanometrisis P.C., 15341 Aghia Paraskevi, Greece; 3Department of Physics, University of Patras, GR 265 00 Patras, Greece; 4Electronic Materials and Devices Research Group, Zepler Institute for Photonics and Nanoelectronics, University of Southampton, Southampton SO171BJ, UK; ak1a12@ecs.soton.ac.uk (A.K.); T.Prodromakis@soton.ac.uk (T.P.)

**Keywords:** Resistive Random Access Memory (ReRAM), Line Edge Roughness (LER), variability, uniformity, modeling, lithography

## Abstract

We investigate the effects of Line Edge Roughness (LER) of electrode lines on the uniformity of Resistive Random Access Memory (ReRAM) device areas in cross-point architectures. To this end, a modeling approach is implemented based on the generation of 2D cross-point patterns with predefined and controlled LER and pattern parameters. The aim is to evaluate the significance of LER in the variability of device areas and their performances and to pinpoint the most critical parameters and conditions. It is found that conventional LER parameters may induce >10% area variability depending on pattern dimensions and cross edge/line correlations. Increased edge correlations in lines such as those that appeared in Double Patterning and Directed Self-assembly Lithography techniques lead to reduced area variability. Finally, a theoretical formula is derived to explain the numerical dependencies of the modeling method.

## 1. Introduction

Resistive Random Access Memory (ReRAM) technologies are considered as one of the most promising candidates for future nonvolatile memory (NVM) applications due to their high memory capacity [1], their simple two terminal architecture, and their excellent scalability [2]. ReRAM cells can be programmed faster than current NVMs and at lower voltages, overall leading to a significant reduction in energy consumption per bit [3]. The cross-point metal-insulator-metal (MIM) cell simple structure requires very low thermal budget and thus can be integrated easily in the current CMOS Back End of Line (BEOL) processing steps [4]. Large memory blocks have been implemented and demonstrated, following the cross-point architecture [5]. The cross-point area F × F, F is the lithography node, can be as small as 2 nm × 2 nm [6] leading to a very small cell footprint (area/bit) 4F^2^. Conductive AFM experiments have demonstrated that the ReRAM memory cell area can be as small as that of an AFM tip [7]. In addition, the 3D stacking of different memory layers [8] reduces the cell’s footprint to 4F^2^/n, where n is the number of the stacked memory layers, and hence severely increases the density of the ReRAM memory chips. Moreover, the ReRAM cell can be used to realize memristive devices to implement new computing paradigms like neuromorphic [3,9], reconfigurable electronics [10], and logic-in-memory [11].

The uniformity of the cell SET and RESET voltages (V_SET_ and V_RESET_, respectively) is a key-challenge to large-scale manufacturing of ReRAM technology. Up to now, it has been mainly related to the stochastic nature of conducting a filament creation process. Remedies that have been proposed consider (a) engineering of electrode/oxide interface [12], (b) inserting seeds (nanoparticles) in the oxide bulk [13], and (c) reduction of the size of cell area [14]. The latter is related to the scaling trends of ReRAM devices and is a great advantage of this technology since scaling down the dimensions of a ReRAM cell goes with the increase in performance uniformity due to the mitigation of stochastic filament formation. Thus, scalability is generally considered an advantage of ReRAM owing to the filamentary conduction and switching mechanisms. The effect of the reduction of the cell size area to the various parameters of ReRAM operation is an active matter of debate. To that end, in Reference [14] a simplified analytical model to describe the area and thickness scaling of forming voltage is proposed. Binary-oxide ReRAM scalability performance has drawn the attention of various groups. The general scaling trend of the high and low resistance states from various metal oxide ReRAMs has been presented in Reference [15]. The dependence of the scaling area on various ReRAM operation parameters (Resistance, SET/RESET current, SET/RESET voltage) has been studied thoroughly for HfO_x_ [16,17], TaO_x_ [18], SiO_2_ [19], ZrO_2_ [20], and recently for Al_2_O_3_ [21] and for TiO_2_ [22]. Similar scalability issues have been examined for perovskite material-based ReRAMs, such as SrTiO_3_ [23], and metal nitrides, such as AlN [24] and NiN [25].

However, up to now no special attention has been paid on deviations frompattern’s uniformity induced by the fabrication process and material properties. In CMOS technology, one of the most important sources of such deviations from manufacturing uniformity is related to the Line Edge Roughness (LER). LER is actually the sidewall roughness of pattern lines manufactured via standard lithographic rules, as often observed via top-down SEM images. The effect of LER becomes more evident as the size of devices is reduced, which is the reason for the recent upsurge of interest in LER control in modern <20 nm semiconductor manufacturing [26]. Obviously, LER is responsible for the area variability of devices in a crossbar (Xbar) array and, furthermore, the cell-to-cell and wafer-to-wafer variability that appear in the operation parameters of ReRAM memory cells in Xbar arrays. More specifically, as mentioned in [15], the resistance at HRS increases as the inverse of the area cell A (i.e., R_HRS_~1/A) while the reset current increases as A decreases. Furthermore, the reduction of the cell size A attributes to local heating causing faster switching times [17]. In addition, forming and SET voltages also increase as the cell area A decreases [17]. These detrimental LER effects are expected to become stronger increasing the size of the memory crossbar array: The larger the memory array the stronger the effects are.

The importance of LER is justified by the evaluation of its effects on device performance. This can be done through carefully designed experiments or modeling/simulation studies. Results from such studies help defining the specifications for allowed LER in roadmaps and factory production lines. Clearly, the level of acceptable LER is defined by the operating constraints of distinct applications that employ nanoscale devices. Up to now, several modeling and experimental investigations have been performed for conventional MOSFET [27,28,29,30] and FinFET [31,32] devices. On the contrary, very few studies have been devoted to the LER effects on memory and especially ReRAM devices. In the fabrication of ReRAM and especially in the cross-point configurations, the LER of metallic lines may affect device performance through the induced variability to the cross-point cell areas and consequently to the device resistance.

Furthermore, it was recently realized that LER characteristics are sensitive to the applied lithographic technique [33]. The main point of differentiation comes from the degree of correlations between the fabricated edges and lines. In EUV patterns, for example, the nearby lines and edges are totally uncorrelated while in Multiple Pattern (MP) techniques correlations are induced mainly between the edges of each line, whereas lines are still independent at least for Double Patterning (DP) schemes. On the contrary, in Directed Self-Assembly (DSA) lithography even nearby lines exhibit correlated fluctuations arising from the very self-assembly nature of the process and the sidewall roughness of grapho-epitaxy grating used for the alignment of lines. Therefore, LER modeling should always consider such aspects and adjust the modeled patterns according to the considered lithography. Up to now, the modeling strategies for the generation of line edges similar to the fabricated ones are based on inverse Fourier techniques, where no correlations are taken into account between edges or lines [34,35].

In order to get a better picture of the LER impact on cell area uniformity, one can observe Figure 1a where several ReRAM prototypes, of cross-point architecture, were fabricated by e-beam lithography and are depicted in a top-down SEM image. Furthermore, on the same SEM image the detected edges defining the borders of electrode lines are illustrated with black color. It can be clearly seen that the roughness of the metal edges induces non-uniform cell areas (see Figure 1b). The distribution of the cell areas is presented in Figure 1c where one can notice the wide distribution of cell area values. This data supports our claim that LER needs to be considered carefully for achieving a uniform and reliable ReRAM technology and deserves more investigation.

In this paper, we evaluate the effects of LER of metallic lines on the device area uniformity, which is critical for the device performance. Three types of line patterns are considered corresponding to EUV, Multiple Patterning (MP), and DSA patterns to compare these with respect to their vulnerability to LER defect. In Section 2, the modeling methodology is presented and explained. The results of the modeling approach to EUV, MP, and DSA-based patterns are shown and discussed in Section 3. In the end of the same section, we also derive an analytical formula capturing the numerical modeling results with quantitative success. The last section, Section 4, summarizes the findings of the paper and draws the main conclusions.

## 2. Modeling Methodology

### 2.1. Background of Modeling: LER Characterization in Different Lithography Techniques

LER is usually defined as the deviation from smoothness (flatness) of the sidewalls of resist or substrate lines. In top-down SEM images, this deviation is reflected through the roughness of the edge defining the 2D borders of the line region. In the initial stages of LER studies, line edge points were considered uncorrelated and the main emphasis was given on the measurement of their root mean square (rms) value (standard deviation), quantifying the wideness of the edge point distribution. However, quite early, the characterization scheme was enriched with more parameters and functions aiming to capture and quantify the spatial/lateral or frequency aspects of LER preserving the dominant significance of rms. To this end, a three-parameter model has been proposed [26] consisting of rms value, correlation length ξ, and roughness exponent α (related to the fractal dimension d = 2−*α*) and extensively used in different applications. The first parameter rms quantifies the vertical aspects of LER and is calculated by the standard deviation of the edge points about their mean value. The other two parameters (ξ and α) focus on the spatial aspects of LER. The correlation length ξ quantifies the window inside which the edge points can be considered correlated, i.e., they have similar deviations from the mean value, and delivers a statistical estimation of the mean width of edge fluctuations. Large values of ξ mean slowly varying edges while small values characterize more jagged edges. The correlation length is normally calculated through the autocorrelation function R(r) as the distance r at which R(r) is lowering beneath 1/e, i.e., R(ξ) = 1/e [36]. The roughness exponent 0 < α < 1 quantifies the contribution of various scales and frequencies to the whole LER. When α takes on low values, high frequencies dominate, whereas whenit approaches 1, the edge becomes smoother at small scales and the importance of high frequency fluctuations is being lessened.

When a more detailed function-like characterization of spatial and frequency LER aspects is demanded, the Power Spectrum (PS) and the Height–Height Correlation Function (HHCF) can be calculated and elaborated. The overall shape of these functions is linked to the three-parameter model, when LER obeys a self-affine fractal symmetry [36]. The effects of LER on line width fluctuations (usually called Line Width Roughness (LWR)) complete the overall picture of the conventional approach to LER metrology. The relationship between LER and LWR metrics has been clarified and it can be shown that the low frequency LWR could have a significant contribution to the total budget of local Critical Dimension Uniformity (CDU). This effect has underlined the importance of estimating the PS of LWR, which quantifies the contribution to LWR from different frequency areas. A detailed presentation of the conventional LER and LWR metrology can be found in [26].

Nevertheless, during the recent years significant changes in lithographic landscape have been evidenced. The conventional scaling down of feature dimensions, based on the optimization of wavelength and Numerical Aperture (NA), has been replaced by a more etch-based and material-driven resolution enhancement, which duplicates the density of patterns through successive deposition and etching steps. The family of these MP techniques is currently used in high volume manufacturing in semiconductor industries. Additionally, the concept of self-assembly of block copolymers has been employed in lithography research in the last 10 years due to its ability to easily provide line/space patterns with widths less than 20 nm. The DSA lithography has seen an upsurge and very important advances have been performed in defect reduction and scaling improvement. However, what is usually missing is the effect that these new lithographic approaches have on LER and its metrology characterization. The key aspect of these effects is the lateral (across line direction) correlations they introduce in line/space patterns. In the EUV lithography patterns, the LER of edges are uncorrelated and no propagation of edge fluctuations is noticed between the edges of a line or the nearby lines of a pattern. On the contrary, in MP lithography patterns, the edges of the same line fluctuate in a correlated manner forming wiggling lines. A more dramatic change is observed in DSA patterns. Here, not only the line edges but also the lines themselves present correlations in the way they fluctuate. To capture these new aspects of LER, an extended framework for LER/LWR metrology has been proposed and elaborated based on the c-factor parameter, function, and correlation length. A detailed presentation and examples of application of these metrics can be found in [35].

### 2.2. Implementation of Modeling

The key idea of our approach is to model LER effects in cross-point area statistics using the 2D projections of the real 3D patterns, as they are shown in top-down SEM images similar to that in Figure 1. The assumption behind this approach is that 3D morphological variations do not have an important contribution to LER impact on device variability. This assumption is supported by recent observations according to which the sidewall roughness of lithographic lines does not change significantly along z-direction since it exhibits a curtain-like morphology [37].

The modeling methodology is implemented in three steps. First, the structural parameters of the cross-point pattern are defined including line width (Critical Dimension, *CD*) and pitch values. The numbers of both vertical and horizontal lines are included in the modeling as well as. These parameters can differ in vertical (y) and horizontal (x) line patterns, though throughout this study we mainly consider the symmetrical case where *CD*_X_ = *CD*_Y_ and pitch_x_ = pitch_y_. Based on these parameters, the ideal smooth pattern can be generated as shown in Figure 2a. In the second step, the smooth edges are converted to rough edges characterized by the input predetermined roughness parameters: rms, ξ, α. The generation of rough edges is made using the convolution method and requires the use of a model function for the Power Spectrum or the Height–Height Correlation Function. Here, we extensively use the following HHCF *G*(*r*) that is also used in self-affine rough processes [34,38]:(1)$$G^{2}\left(r\right)=2rms^{2}\left(1-e^{{\left(-\frac{r}{\xi }\right)}^{2\alpha }}\right).$$

The successive application of the convolution method with no further processing of edges produces uncorrelated edges, which then can be positioned according to *CD* and pitch values to build the whole cross-point pattern. The generated pattern can be used to model the 2D projection of line/space structures fabricated by mask-based lithographic rules (EUV or 193i). When MP or DSA patterns are sought, we may add correlation either between the edges of the same line or to expand these between nearby lines to get DSA-like patterns. Figure 2b shows examples of such patterns with controlled LER parameters as well as cross-edge and cross-line correlations.

In the third step of our methodology, we measure the areas of cross-points A_i_, I = 1, …, N_x_N_y_ where horizontal and vertical lines overlap in the considered rough pattern of N_x_xN_y_ lines. Then we calculate the normalized standard deviation *σ*(*Α*) of the A_i_ values (=*std*(*A_i_*)/< *A_i_* >) and we plot it for various dimensional and LER parameters and edge/line correlation levels (see Figure 2c).

## 3. Results and Discussion

The results of our modeling calculations are collectively illustrated in Figure 3, which displays six contour plots. Each one of these plots depicts the dependence of *σ*(*A*) on the LER parameters rms and ξ given that these parameters are considered to quantify LER according to the last editions of ITRS [39]. In all runs, the roughness exponent α is kept fixed to 0.5 in conformity with more experimental measurements. We consider that both horizontal and vertical lines have similar LER parameters. The left column of Figure 3a,c,e contains the contour plots with *CD* = 20 nm and pitch = 40 nm while the right column shows the results when the pattern scales down at *CD* = 10 nm and pitch = 20 nm. The different rows account for increased levels of horizontal (cross-line) correlations between edges and lines. In the first row, both edges and lines are generated to be uncorrelated to capture the case of mask lithography line structures (EUV and 193i). The second row contains the results for cross-point patterns where the edges of each line are correlated, though the lines themselves fluctuate independently. The latter pattern resembles the line/space structures acquired by Double Patterning Lithography techniques. In the third row, the contour plots concern model patterns where the horizontal correlations propagate across both edges and lines similarly to what is happening in DSAL.

From the contour plots of Figure 3, we are able to extract conclusions and discuss the following issues: (a) the dependence of *σ*(*A*) on LER parameters rms and ξ which seems to exhibit a similar pattern in all cases, (b) the effects of scaling down the pattern dimensions, and (c) the effects of cross-edge and line correlations.

### 3.1. The Impact of LER Parameters Rms, ξ

The overall shape of all contour plots in Figure 3 is characterized by the dominance of vertical contour lines, which reveal the stronger dependence of area uniformity on rms with respect to that of the correlation length. In order to focus on the dependencies themselves, we take a horizontal and vertical cross-section of the contour diagrams and the outcome plots are shown in Figure 4 for all considered cases. One can easily notice the linear increase of *σ*(*Α*) with rms value in all cases, while the effects of ξ are sublinear justifying the dominant role of rms in the effects of LER on cross area uniformity. In other words, both rms and ξ in LER degrade device area variability with rms exhibiting the primary effect.

### 3.2. The Impact of Scaling Device Size

The comparison of the contour plots of the left and right columns reveals that the scaling down of pattern dimensions by a factor 2 (quadrupling of device density) is associated with an almost doubling of normalized area variability *σ*(*Α*). This observation holds at all rows, i.e., it is independent of the level of edge/line correlations.

### 3.3. The Impact of Edge and Line Correlations

When MP or DSA lithographic techniques are used for ReRAM pattern fabrication, edges and lines exhibit cross-correlations, i.e., they fluctuate in a correlated manner. The second and third rows of Figure 3 show the contour plots for *σ*(*Α*) when the pattern exhibits only edge and both edge and line correlations, respectively. Although the pattern of contour lines remain almost unaltered, the *σ*(*Α*) values demonstrate a clear reduction with respect to the values of first rows approximately by a factor 3. The amount of reduction is similar at the contour plots of the second and third rows, which indicates that the critical correlations for variability drop are those between edges. The cross-line correlations do not seem to play any role in LER-induced degradation effects on area uniformity.

### 3.4. Analytical Formula for LER Effects on Area Variability

The dependencies of *σ*(*A*) on LER parameters found by modeling and shown in Figure 3 can be captured in an analytical formula which can be derived as follows: For patterns with smooth edges (no LER) the cross-point areas are fixed in all devices and equal to *A_i_* = *CD*_X_*CD*_Y_ with no variability at all. When we consider rough edges, LER induces local variability in line widths, which is usually called Line Width Roughness (LWR). Therefore, each crossing area can be roughly characterized by A_i_ = *CD*_ix_*CD*_iy_, where *CD*_ix_ and *CD*_iy_ are the average line widths across x- and y-directions enclosed in the i-th crossing area. Due to LER and LWR, *CD*_ix_ and *CD*_iy_ change randomly from area to area. If we assume that on average this variability is similar to both vertical and horizontal lines, we get for the *σ*(*Α*):(2)$$\sigma \left(Α\right)=\frac{std\left(A_{i}\right)}{\langle A_{i}\rangle }\approx \frac{\surd 2std\left(CD_{i}\right)}{\langle CD_{i}\rangle }.$$

The relationship of *CD* variance (*std*^2^(*CD_i_*)) with local and total variances of LWR values has been studied extensively in LER literature and the following formula can be proven [20]:(3)$$std^{2}\left(CD_{i}\right)+<rms_{LWR}^{2}\left(CD_{i}\right)>=\;rms_{LWR}^{2}\left(total\right),$$where $rms_{LWR}^{}\left(total\right)$ is the rms value of LWR for the total lines included in model patterns, which assume to get sufficiently large lengths so that the *rms_LWR_* value stabilizes at a fixed value, and $<rms_{LWR}^{2}\left(CD_{i}\right)>$ is the mean variance of LWR calculated inside the segments of length *CD_i_* and averaged over all considered segments.

In order to get the relationship of *σ*(*Α*) with the input LER parameters, we need to pass from LWR to LER parameters. This can be achieved through the following formula connecting LWR and LER assuming similarity between left and right LER:(4)$$rms_{LWR}^{2}=2\left(1-c\right)\;rms_{LER}^{2}.$$

Here, *c* is the c-factor quantifying the cross-correlations between the left and right edges of lines. For totally uncorrelated edges, *c* = 0, whereas for fully correlated (anti-correlated), *c* = 1 (−1) [26,33]. By combining (2), (3), and (4) we read:(5)$$\sigma \left(Α\right)=\frac{2}{\langle CD_{i}\rangle }\;\sqrt{\left(1-c\right)(rms_{LER}^{2}\left(total\right)-<rms_{LER}^{2}\left(CD_{i})>\right)}.$$

The Formula (5) can be further processed in the case of edges with exponential autocorrelation functions (roughness exponent *α* = 0.5) since for such edges it holds that

(6)$$rms_{LER\;}^{2}\left(CD\right)=rms_{LER}^{2}\left(total\right)[1-\frac{2\xi }{CD}(1+\frac{\xi }{CD}e^{-\frac{CD}{\xi }}-1)].$$

Inserting (6) into (5) we get the final formula for the dependence of *σ*(*A*) on LER parameters:(7)$$\sigma \left(Α\right)=2\sqrt{2\left(1-c\right)}\;\frac{rms_{LER\;}}{CD}\frac{\xi }{CD}\sqrt{\frac{CD}{\xi }+e^{-CD/\xi }-1}.$$

Equation (7) incorporates all the dependencies of normalized area variability on the dimensional parameters (*CD*), the principal LER parameters (*rms_LER_*, ξ) and the level of edge correlations (c-factor). As expected from the numerical results, pitch and line correlations are not included since they have no impact. One can easily notice that the analytical Formula (7) predicts (a) the linear proportionality of *σ*(*Α*) to *rms_LER_*, (b) the sublinear increase of *σ*(*Α*) with ξ, (c) the almost linear inverse proportionality impact of *CD*, and (d) the reduced effects of increased edge correlations on *σ*(*Α*) quantified by *c* through the factor (1-*c*).

In order to get a more quantified and complete comparison of the analytical prediction with the numerical results, we show in Figure 5a a contour plot with the dependencies of *σ*(*Α*) on LER parameters (rms, ξ) as derived by the analytical Formula (7). We have chosen *c* = 0 (no edge correlations) and *CD* = 20 nm, which are the parameters of the numerical contour plot of Figure 3a. One can clearly indicate the striking qualitative and quantitative similarity of two contour plots, which is more directly and quantitatively illustrated in Figure 5b where the difference of the numerical and analytical normalized variability *σ*(*Α*) is shown versus rms and ξ parameters. The difference is always very close to zero and justifies the predictive power of the analytical formula. The sole exception is for the small rms values lower than 0.5 nm where the analytical formula seems to underestimate the numerical results. However, the observed discrepancy is due to issues with the numerical results. More specifically, in order to simulate the real procedure where the LER effects are evaluated through the analysis of SEM images (see Figure 1), in the numerical modeling we pixelize the edge data by rounding the generated edge points from the generator algorithms. We have shown that this discretization (pixelization) process induces a noise impact on LER and causes an increase of the measured rms value as explained in [40]. Therefore, in these regimes with extremely small rms values, one should trust the predictions of the analytical Formula (7) more than the numerical results of the 2D modeling.

The similarity between numerical results and analytical predictions was confirmed independently of *CD* and edge correlations. This means that the Formula (7) comprises a strong analytical tool to get control and deeper understanding of the LER effects on ReRAM area variability and performance degradation since it captures successfully the impact of LER main parameters (rms, ξ) as well as of the line width (*CD*) and edge correlations which may come from different lithography steps used in pattern formation.

## 4. Conclusions

In this work, we model the effect of lithographic performance parameters on the operation of ReRAM devices, assuming the main three lithographic techniques: EUV, MP and DSA. According to our numerical model, it is found that both rms and ξ in LER degrade device area variability with rms exhibiting the primary effect. Additionally, it is found that edge correlations favor area uniformity while the normalized area variability is doubled when the size becomes half of the initial. For comparison, an analytical model was developed also to simulate the impact of lithography performance parameters. The comparison between the 2D numerical and the analytical one reveals that both provide the same results independently of *CD* and edge correlations. Nevertheless, the analytical model is more accurate at extremely small rms values. In this context, the developed numerical and analytical models are highly valuable predictive tools for the integration of ReRAM devices at very small technology nodes.

## Figures and Tables

**Figure 1 materials-12-03972-f001:**
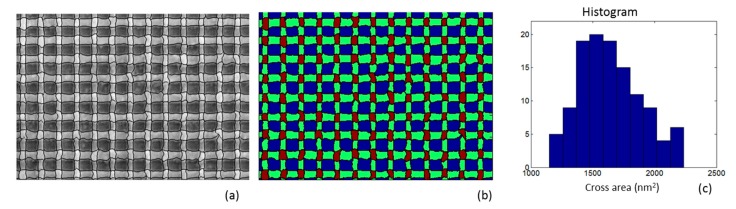
(**a**) A top-down SEM image of a cross-point pattern with the detected edges of the vertical and horizontal metal lines defining the crossing areas, (**b**) a colored illustration of the SEM image to show the crossing areas more clearly in red color and the metal lines in green color, (**c**) the histogram of the calculated cross-point area values in nm^2^. The Line Edge Roughness (LER) (average root mean square (rms) value) of the vertical and horizontal lines was measured and found equal to 2.6 and 3.8 nm, respectively. The width of the distribution of cross areas (standard deviation) is found to be almost 16% of the mean value. Therefore, the commonly used three times this value reaches almost 50% of the mean value indicating the strong impact of LER of metal lines to the nonuniformity of Resistive Random Access Memory (ReRAM) device areas.

**Figure 2 materials-12-03972-f002:**
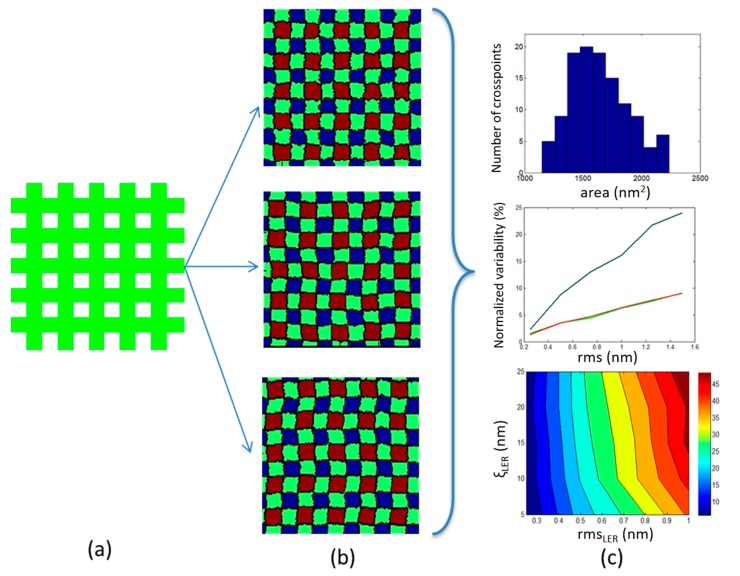
A schematic view of our modeling steps for LER effects on device uniformity. First, a pattern is formed (**a**) with the given parameters (Critical Dimension (*CD*), pitch, number of lines), then (**b**) roughness is imposed on the edges to get LER with predefined parameters, and then in (**c**) the statistical analysis of cross areas A_i_ is performed to get the normalized standard deviation *σ*(*Α*) versus LER parameters.

**Figure 3 materials-12-03972-f003:**
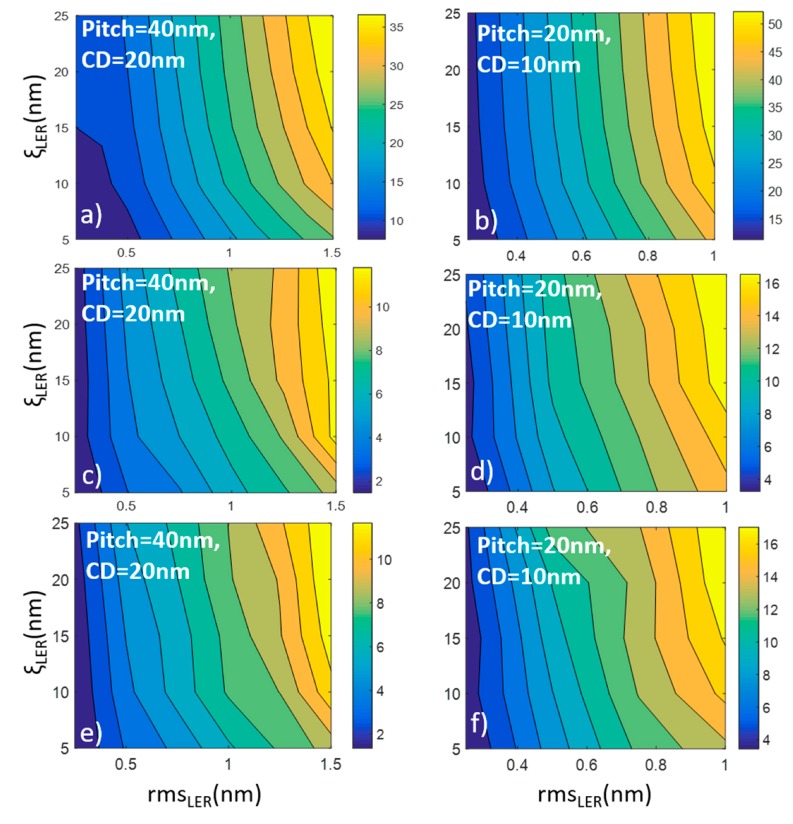
Contour plots of *σ*(*A*) vs. rms_LER_ and correlation length ξ_LER_. Left (**a**,**c**,**e**) and right (**b**,**d**,**f**) columns contain the plots for *CD* = 20 nm and *CD* = 10 nm, respectively. Furthermore, in the second (**c**,**d**) and third (**e**,**f**) row the calculations have been done considering patterns with correlated edges and correlated edges and lines respectively.

**Figure 4 materials-12-03972-f004:**
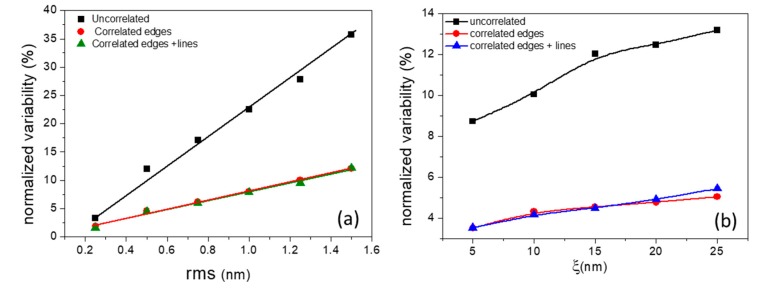
An example of the dependence of the normalized variability *σ*(*A*) on (**a**) rms and (**b**) ξ for typical parameters. One can notice the linear and sublinear increase of *σ*(*Α*) vs. rms and ξ, respectively.

**Figure 5 materials-12-03972-f005:**
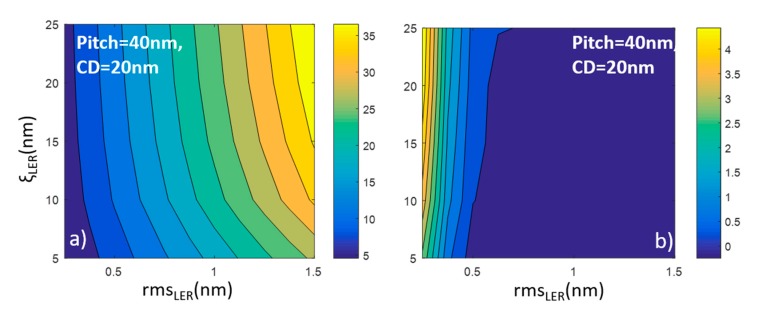
(**a**) Contour plot of the normalized variability *σ*(*Α*) cross-point areas versus the LER parameters rms and ξ for *CD* = 20 nm and *c* = 0 (no edge correlations) and (**b**) the difference of numerical and analytical predictions for *σ*(*Α*) for the standard parameters *CD* = 20 nm and *c* = 0. One can notice the striking similarity of numerical and analytical results with the slight exception of very small rms values where numerical results are biased due to pixelization of edge data.

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
