# Peer review of "Impact of Line Edge Roughness on ReRAM Uniformity and Scaling"

_materials, 2019, doi:10.3390/ma12233972_

Round 1

Reviewer 1 Report

The authors explore effects of the LER on the variability of junction area of ReRAM. The Problems of LER have been mostly discussed focusing on the channel length variability of MOSFET and width variability of FinFET body as LWR (Line Width Roughness). Although the area variability can be deduced from the analysis of LWR, effects of LER on the junction area of cross-point ReRAM is an future issue. In the manuscript, authors describe the variability of area by the simple analytical formula. The reviewer thinks that the paper should be appeared in MDPI Materials after some revisions.

Problems:

There are many small mistakes. Authors should inspect the manuscript more precisely. Followings are some of the mistakes I found.

Line 100: DSA is defined again differently. Line 167: “Fig. 3a” should be “Fig. 2a”, and Line 183: Figure 3b should be Figure 2b. Fig. 2(c) is not referred in the main body of the text. In Fig. 2(c), there is no explanation for horizontal and vertical axes of the three graphs. In the caption of Fig. 2, there is no explanation for (c). Line 201: DPL is not defined.

Author Response

Dear Editor,

First of all, we would like to thank the reviewers of our ms for their comments and criticisms. Below could you find our responses to the comments of reviewer 1 in red along with references to the modifications of the text in the submitted revised ms.  We believe that now our work is presented in a better and clearer way thanks to reviewers’ comments.

Best regards,

Vassilis Constantoudis

REVIEWER 1

Comments and Suggestions for Authors

The authors explore effects of the LER on the variability of junction area of ReRAM. The Problems of LER have been mostly discussed focusing on the channel length variability of MOSFET and width variability of FinFET body as LWR (Line Width Roughness). Although the area variability can be deduced from the analysis of LWR, effects of LER on the junction area of cross-point ReRAM is an future issue. In the manuscript, authors describe the variability of area by the simple analytical formula. The reviewer thinks that the paper should be appeared in MDPIMaterials after some revisions.

Problems:

There are many small mistakes. Authors should inspect the manuscript more precisely. Followings are some of the mistakes I found.

Line 100 (108 in revised ms): DSA is defined again differently.

Answer: Thanks for noticing it. We have made the correction and now DSA is defined at Line 83 as Directed Self-Assembly.

Line 167 (185 in revised ms): “Fig. 3a” should be “Fig. 2a”, and Line 183 (202 in revised ms): Figure 3b should be Figure 2b. Fig. 2(c) is not referred in the main body of the text.

Answer: We have made the suggested corrections and added a reference to Fig. 2c at line 207.

In Fig. 2(c), there is no explanation for horizontal and vertical axes of the three graphs.

Answer: We have added the axes titles in all graphs of Fig. 2c. Thanks!

In the caption of Fig. 2, there is no explanation for (c).

Answer: We have added it. Thanks!!

Line 201 (220 in revised ms): DPL is not defined. 

Answer: DPL stands for Double Patterning Lithography. We have defined it in 220-221

Reviewer 2 Report

The manuscript is trying to explore the impact of line edge roughness on ReRAM uniformity and scaling through modeling and results analysis. The contents are well written, however, there are still several technical issues regarding the details of the work. Thus the manuscript has to be revised before considering as a potential publication in Materials. Here list some points for the authors to think through.

The SEM pictures in Figs. 1 showed inconsistency of metal lines at the junctions. Why? Is there any inter-dielectrics between active cells for this study? the correlation length parameter should be further illustrated such as how it is defined and so on. In page 5 and 6, Figs. 3 are referred. However, it might be Figs. 2, otherwise Figs. 2 would not be referred. In equation 1, what is r stands for? If it is rms, then token should be consistent throughout the same formula, please correct correspondingly. In addition, how is this equation been used during modeling. Please provide some more details. For the entire work, the authors have been trying to explore the geometrical variations or non-uniformities of the ReRAM scaling by building the models. However, if these geometrical non-uniformities would not extensively impact ReRAM's electrical behavior, then the work will make no sense at all. It is highly recommended in some part of the manuscript to add discussions of such relationships. How is the simulation results of LER or area variations would further impact ReRAM electrical behaviors. Such work would make more contributions to the whole ReRAM community.

Author Response

Dear Editor,

First of all, we would like to thank the reviewers of our ms for their comments and criticisms. Below could you find our responses to the comments of reviewer 2 in red along with references to the modifications of the text in the submitted revised ms.  We believe that now our work is presented in a better and clearer way thanks to reviewers’ comments.

Best regards,

Vassilis Constantoudis

REVIEWER 2

Comments and Suggestions for Authors

The manuscript is trying to explore the impact of line edge roughness on ReRAM uniformity and scaling through modeling and results analysis. The contents are well written, however, there are still several technical issues regarding the details of the work. Thus the manuscript has to be revised before considering as a potential publication in Materials. Here list some points for the authors to think through.

The SEM pictures in Figs. 1 showed inconsistency of metal lines at the junctions. Why? Is there any inter-dielectrics between active cells for this study?

Answer: We have specifically included the SEM picture shown in Fig,1 for illustrating the adverse impact that the LER of lines may have on the variability of cross-point areas, as also described in [Adam et al., “Challenges hindering memristive neuromorphic hardware from going mainstream” Nature Communications (2018) 9:5267]. This issue is accentuated as critical dimensions reach the deep sub-micron regime (~10nm). At that scale there are many reasons leading to such LER effects, related with lift-off and/or poor adhesion of metal lines on the underpinning layer. In this particular case, a TiO2 thin layer was employed between the bottom and top electrodes. But this merely serves as only one possible example as the aim of the paper is to focus on the methodology behind studying LER and its impact rather a specific case of use.

The correlation length parameter should be further illustrated such as how it is defined and so on.

Answer: We have provided more info about all LER parameters with more emphasis on the definition and calculation of the correlation length (see lines 133-143)

“The first parameter rms quantifies the vertical aspects of LER and is calculated by the standard deviation of the edge points about their mean value. The other two parameters (ξ and α) focus on the spatial aspects of LER. The correlation length ξ quantifies the window inside which the edge points can be considered correlated, i.e. they have similar deviations from the mean value, and delivers a statistical estimation of the mean width of edge fluctuations. Large values of ξ mean slowly varying edges while small values characterise more jagged edges. The correlation length is normally calculated through the autocorrelation function R(r) as the distance r at which R(r) is lowering beneath 1/e, i.e. R(ξ)=1/e [36].  The roughness exponent 0<α<1 quantifies the contribution of various scales and frequencies to the whole LER. When α takes on low values, high frequencies dominate whereas as it approaches 1 edge becomes smoother at small scales and the importance of high frequency fluctuations is being lessened.”

In page 5 and 6, Figs. 3 are referred. However, it might be Figs. 2, otherwise Figs. 2 would not be referred.

Answer: We have updated the references to the Figures 2,3 according to the reviewer’s comment. Thanks!! (see also point 2 by Reviewer 1)

In equation 1, what is r stands for? If it is rms, then token should be consistent throughout the same formula, please correct correspondingly. In addition, how is this equation been used during modeling. Please provide some more details.

Answer: In equation (1), r stands for the distance between two edge points along their direction as it is common in the definition of the correlation functions of all types.

The formula of G(r) (and similarly of R(r)) is used to estimate the corresponding PSD which is then multiplied by the PSD of white noise to randomize phases. The inverse Fourier transform of the PSD product gives the desired edge with LER determined by the parameters in the input formula (1).

For the entire work, the authors have been trying to explore the geometrical variations or non-uniformities of the ReRAM scaling by building the models. However, if these geometrical non-uniformities would not extensively impact ReRAM's electrical behavior, then the work will make no sense at all. It is highly recommended in some part of the manuscript to add discussions of such relationships. How is the simulation results of LER or area variations would further impact ReRAM electrical behaviors. Such work would make more contributions to the whole ReRAM community. 

Answer: The reviewer is right because we did not emphasize the LER effects on ReRAM operations. In the introduction, lines 43-76, we cited the most important published articles related to the effect of the ReRAM cell area variation on the operational characteristics of the memory cell, such as the resistance, the SET/RESET current and SET/RESET voltage. Specifically, we cited research articles where the area dependence is revealed in ReRAM cell with different resistance switching materials. In order to emphasize the LER effects on ReRAM cells, we add the following text (lines 67-75):

Obviously, LER is responsible for the area variability of devices in a crossbar (Xbar) array and furthermore the cell-to-cell and wafer-to-wafer variability that appear in the operation parameters of ReRAM memory cells in Xbar arrays. More specifically, as mentioned in [15], it was found that the resistance at HRS increases as the inverse of the area cell A (i.e. RHRS ~ 1/A) while the reset current increases as A decreases. Furthermore, in the reduction of the cell size A attributes to local heating causing faster switching times [17]. In addition, forming and SET voltages also increase as the cell area A decreases [17]. These detrimental LER effects is expected to become stronger increasing the size of the memory crossbar array: the larger the memory array the stronger the effects are.”

Round 2

Reviewer 2 Report

The authors have address all technical concerns in the previous version manuscript and I do not have any other concerns. 

Thus I would suggest acceptance of the manuscript.